# Blood–Brain Barrier Disruption (BBBD)-Based Immunochemotherapy for Primary Central Nervous System Lymphoma (PCNSL), Early Results of a Phase II Study

**DOI:** 10.3390/cancers15041341

**Published:** 2023-02-20

**Authors:** Hanne K. Kuitunen, Aino L. K. Rönkä, Eila M. Sonkajärvi, Juha-Matti Isokangas, Marja Pyörälä, Kari A. A. Palosaari, Anna S. Jokimäki, Anu E. Partanen, Harri J. Littow, Merja A. Vakkala, Esa J. Jantunen, Mirja E. Huttunen, Katja J. Marin, Annikki M. K. Aromaa-Häyhä, Päivi K. Auvinen, Tuomas Selander, Inka K. Puhakka, Outi M. Kuittinen

**Affiliations:** 1Cancer Center, Oulu University Hospital, 90220 Oulu, Finland; 2Department of Oncology and Radiotherapy, Kuopio University Hospital, 70210 Kuopio, Finland; 3Surgery and Anaesthesia Center, Oulu University Hospital, 90220 Oulu, Finland; 4Service for Medical Care, Oulu University Hospital Diagnostics, 90220 Oulu, Finland; 5Department of Medicine, Kuopio University Hospital, 70210 Kuopio, Finland; 6Medical Research Center Oulu, Research Group of Surgery, Anesthesiology and Intensive Care Medicine, 90220 Oulu, Finland; 7Institute of Clinical Medicine, University of Eastern Finland and Department of Medicine, 70210 Kuopio, Finland; 8Hospital District of North Carelia, Joensuu Central Hospital, 80210 Joensuu, Finland; 9Science Service Center, Kuopio University Hospital, 70210 Kuopio, Finland; 10Department of Neurology, Kuopio University Hospital, 70210 Kuopio, Finland; 11School of Medicine, Institute of Clinical Medicine, Oncology, Faculty of Medicine, University of Eastern Finland, 70210 Kuopio, Finland

**Keywords:** BBBD, PCNSL, ASCT

## Abstract

**Simple Summary:**

We present the results of a prospective trial of systemic chemoimmunotherapy for primary CNS lymphoma including the intra-arterial administration of methotrexate and carboplatin with blood–brain barrier disruption. The safety, response rates, and survival outcomes were favorable. Previous reports of the intra-arterial administration of therapy with blood–brain barrier disruption for this disease have also demonstrated safety and favorable outcomes, but there has been a slow uptake of this approach, partially out of concern that the technique may be difficult to implement across centers. Our data provide further prospective evidence that this approach can be implemented elsewhere with safety and confirm the effectiveness previously reported in a single institution study. These data motivate further study of this approach not only for PCNSL but also for other diseases that exist behind the blood–brain barrier.

**Abstract:**

Primary central nervous system lymphoma is a rare but aggressive brain malignancy. It is associated with poor prognosis even with the current standard of care. The aim of this study was to evaluate the effect and tolerability of blood–brain barrier disruption treatment combined with high-dose treatment with autologous stem cell transplantation as consolidation on primary central nervous system lymphoma patients. We performed a prospective phase II study for 25 patients with previously untreated primary central nervous system lymphoma. The blood–brain barrier disruption treatment was initiated 3–4 weeks after the MATRix regimen using the previously optimized therapy protocol. Briefly, each chemotherapy cycle included two subsequent intra-arterial blood–brain barrier disruption treatments on days 1 and 2 via either one of the internal carotid arteries or vertebral arteries. Patients received the therapy in 3-week intervals. The treatment was continued for two more courses after achieving a maximal radiological response to the maximum of six courses. The complete treatment response was observed in 88.0% of the patients. At the median follow-up time of 30 months, median progression-free and overall survivals were not reached. The 2-year overall and progression-free survival rates were 67.1% and 70.3%, respectively. Blood–brain barrier disruption treatment is a promising option for primary central nervous system lymphoma with an acceptable toxicity profile.

## 1. Introduction

Primary central nervous system (CNS) lymphoma (PCNSL) is a rare malignancy accounting for approximately 4% of all brain neoplasms; however, the incidence appears to be increasing [1,2]. Histologically, almost all cases present with a diffuse large B-cell lymphoma (DLBCL) histology [3].

The conventional treatment of PCNSL is high-dose (HD) methotrexate-based combination chemotherapy with either whole-brain radiation therapy (WBRT) or HD chemotherapy with autologous stem cell transplantation (ASCT) as consolidation [4,5,6,7,8,9,10,11,12,13]. In a prospective randomized phase II study by the International Extranodal Lymphoma Study Group (IELSG), the best therapy outcomes were achieved with the MATRix regimen (chemoimmunotherapy with methotrexate, cytarabine, thiotepa, and rituximab), with a 2-year progression-free survival (PFS) rate of 61% [14]. In the same trial, a second randomization compared the HD + ASCT treatment with WBRT as a consolidation therapy [15]. Both consolidation options offered similar survival outcomes at the expense of a neurocognitive decline in the radiation therapy arm [15]. Therefore, induction with the MATRix regimen followed by HD + ASCT consolidation should be regarded as the current standard of care for PCNSL [15]. PCNSL relapses may, however, occur even >10 years after primary therapy, and the number of long-term survivors seems to be limited with conventional approaches, establishing a need for novel therapy options [16].

A major challenge in the treatment of CNS malignancies is the fact that the blood–brain barrier (BBB) prohibits the entrance of many effective chemotherapeutic agents into the CNS [17]. Blood–brain barrier disruption (BBBD), developed by Neuwelt et al. is a method to tackle this issue [17]. In BBBD treatment, the BBB is temporarily opened with an intra-arterial infusion of hypertonic mannitol solution [17]. Chemotherapeutic agents are subsequently infused directly to the cerebral arteries, enabling considerably higher drug concentrations in the CNS than in the conventional intravenous therapy protocol [17]. Furthermore, the approach allows the use of several effective lymphoma chemotherapeutic agents that cannot normally pass through the intact BBB [17].

Previously, promising BBBD treatment outcomes with durable disease control have been reported by Angelov et al. [18]. In the study, 149 newly diagnosed PCNSL patients were treated with 12 chemotherapy courses without consolidation therapy with BBBD and intra-arterial (IA) methotrexate. The study demonstrated a good safety profile and neurocognitive tolerance. The overall response rate was 81.9% and median overall survival was 3.1 years [18]. We launched the BBBD protocol in 2007 in Oulu University Hospital, Finland [19]. After gaining treatment experience, we found the original protocol was highly tolerated [19]. However, a considerable number of patients had insufficient responses especially in the relapsed setting, and for this reason we intensified the original protocol from a combination of four to five chemotherapeutic agents administered in an interval shortened from four to three weeks [18,19]. We also decreased the number of BBBD courses into four to five based on the individual timing of the response (every arterial trunk treated twice after achieving a maximal response) and added components of the conventional IELSG protocol (a cytoreductive MATRix induction cycle and an HD + ASCT consolidation) to the treatment [14,15,19]. In our previous retrospective analysis, BBBD treatment with these adjustments showed a highly promising prognosis in patients with previously untreated PCNSL [19]. The complete response rate was 100%, with 2-year progression-free and overall survival rates of 100% [19].

Here, we report the results from a prospective phase II trial analyzing the efficacy and safety of the modified BBBD treatment in combination with chemoimmunotherapy and ASCT-supported HD therapy consolidation in patients with previously untreated PCNSL.

## 2. Materials and Methods

### 2.1. Study Design and Participants

This is a prospective two-arm phase II study evaluating the efficacy and toxicity of BBBD treatment in conjunction with chemoimmunotherapy and ASCT-supported HD therapy consolidation in patients with newly diagnosed (first-line arm) or relapsed/refractory (R/R arm) PCNSL. The recruitment goal was 25 patients in both arms. In the first-line arm, the goal was achieved in November 2020, and the results are presented here. Recruitment is still ongoing for the R/R arm, and the results will be presented later.

To be eligible for the trial, participants had to be 18–70 years old, with an Eastern Cooperative Oncology Group (ECOG) performance status of <2 (unless due to lymphoma), and a newly diagnosed PCNSL (first-line arm) or an R/R PCNSL after conventional systemic chemo(immuno)therapy (R/R arm). Either a biopsy-proven histological diagnosis or cerebrospinal fluid (CSF)/aqueous cytological diagnosis of DLBCL and written informed consent were required.

The following staging work-up and pretreatment evaluation were conducted before the start of the treatment: physical examination, contrast-enhanced whole-brain magnetic resonance imaging (MRI) (within 2 weeks), contrast-enhanced whole-body computed tomography (CT), CSF examination (cytological examination, physicochemical examination, and flow cytometry), ophthalmological assessment (including slit-lamp examination), audiogram, and neuropsychological examination by neuropsychologists.

The medical history of the participants was reviewed by an anesthesiology specialist to evaluate their suitability for the BBBD treatment. Cardiac ultrasound was performed if necessary.

All patients were immunocompetent with no history of immunosuppressive medication and negative HIV, HBV, and HCV serologies.

Risk groups were defined according to the Memorial Sloan–Kettering Cancer Center (MSKCC) prognostic scoring system [20].

This study was registered to the EU Clinical Trials Register, with a EudraCT number of 2014-005015-16, and approved by the Finnish National Ethics Review Board (70/06.00.01/2016).

### 2.2. Outcomes

The primary endpoint was overall survival (OS) rate at the 2-, 5-, and 10-year follow-up. OS was calculated from the date of the chemotherapy initiation to the date of death from any cause.

The secondary endpoints were partial and complete response rates; PFS; time-to-progression (TTP) rates and freedom-from-progression (FFP) at the 2-, 5-, and 10-year follow-up; and treatment-related toxicities. PFS was calculated from the date of chemotherapy initiation to the date of last follow-up, lymphoma progression, or death, whichever occurred first. TTP was calculated from the date of chemotherapy initiation to the date of last follow-up or lymphoma progression, whichever occurred first. Freedom-from-progression (FFP) meant the proportion of patients whose disease was not progressing at the selected time, and patients who died due to other causes were censored.

### 2.3. Procedures

For debulking, patients first received one course of intravenous immunochemotherapy with the MATRix regimen [14].

The BBBD treatment was initiated 3–4 weeks after the MATRix regimen using the previously optimized [19] therapy protocol (Table 1). Briefly, each chemotherapy cycle included two subsequent intra-arterial BBBD treatments on days 1 and 2 via either one of the internal carotid arteries or vertebral arteries. In the treatment schedule prior to the first or second treatment course, a porta-cath was inserted, and patients were hydrated at 100–150 mL/h for a minimum of 6 h. Because of the risk of seizures, patients were premedicated with an anticonvulsant. Rituximab was administered on day 0. On day 1 and 2, Atropine was administered intravenously immediately prior to BBBD to prevent bradycardia. BBB opening was conducted under general anesthesia. The internal carotid artery (ICA) or vertebral artery (VA) was selectively catheterized via transfemoral access. In the ICA, the catheter tip was placed at C1–C2 level and in the vertebral artery at C6 level. Cyclophoshamide and etoposide are pro-drugs and activate liver metabolism and were administered intravenously before mannitol infusion. Warmed mannitol (25%) was administered at 4–6 mL/s into the target, and immediately after BBBD, methotrexate and carboplatin were administered intra-arterially. Following BBBD, patients remained in the post-anesthesia care unit with frequent monitoring of vital signs, neurologic status, and fluid balance. Fluid balance was meticulously maintained with mannitol (15%) and fluid boluses. In patients treated with methotrexate, NaHCO3 was added to intravenous fluids and titrated to achieve urine pH greater than 7.5. Patients received the therapy in 3-week intervals. The treatment was continued for two more courses after achieving the maximal radiological response to the maximum of six courses.

For the ASCT-supported HD chemotherapy, CD34+ stem cells were mobilized with one of the BBBD treatment cycles (cycles III–VI) in patients without progressive disease and harvested as previously described [19]. The collection target level was at least 4.0 × 10^6^ CD34 positive cells/kg of body weight. Plerixafor was used, if necessary, to boost the mobilization of CD34+ cells. The HD therapy protocol included BCNU 400 mg/m^2^ on d-6 and thiotepa 5 mg/kg twice a day (d-5 to -4) [14,15,21].

### 2.4. Local Treatment of Intraocular Lymphoma

Patients with intraocular lymphoma were treated with intravitreal methotrexate and rituximab injections as described by Yeh and Wilson [22].

### 2.5. Side Effects

Treatment-related side effects were assessed and graded according to the Common Terminology Criteria for Adverse Events version 4.03 after each BBBD treatment cycle and after the HD + ASCT treatment. Toxicities from the BBBD treatment and the HD + ASCT treatment were evaluated separately. For each adverse event (AE) during the BBBD treatment, the highest-grade toxicity observed per patient was reported. Neurological toxicities were of special interest in the trial. Acute neurological toxicities were reported in the current study. Chronic neurological toxicities, including the effect of the treatment on neurocognitive functions, assessed with neuropsychological tests, will be reported later.

### 2.6. Response Assessment

The treatment response was assessed using contrast-enhanced brain MRI after each chemotherapy course. Clinical and radiological restaging was performed 1 month after the end of the HD chemotherapy treatment; subsequently, follow-up visits, including radiological assessments, were organized every 3 months up to 2 years, and once a year thereafter for 10 years.

The treatment response was evaluated following the International Primary CNS Lymphoma Collaborative Group (IPCG) response criteria [20]. Briefly, progressive disease was defined as a >25% increase in tumor size or the appearance of a new tumor lesion. Partial response was defined as >50% decrease in tumor size. Complete remission was defined as the complete disappearance of all lymphoma lesions in T1-weighted MRI, negative CSF cytology, and no evidence of disease in the ophthalmological examination. In the case of minor slowly regressing MRI lesions defined as unconfirmed complete remission (CRu) in the IPCG criteria, we performed a positron emission tomography (PET) scan to assess their metabolic state. PET-negative CRu lesions were considered as a complete response.

### 2.7. Statistics

Basic demographics were expressed as means and standard deviations or frequencies and percentages. The Kaplan–Meier method was used to estimate survival rates with 95% confidence intervals (CIs) for outcome variables. The statistics were performed using IBM SPSS software (IBM SPSS Statistics for Windows, Version 26.0, IBM Corporation, Armonk, NY, USA) and R statistical software, version 4.0.4.

## 3. Results

A total of 25 patients were recruited between 4 April 2017 and 10 November 2020. Patient demographics and MSKCC risk groups are presented in Table 2 [23].

A mean of 4.24 (±0.355, 95% CI) and maximum of 6 BBBD treatment cycles were administered per patient. The BBBD treatment was interrupted in two patients after two and three cycles because of unforeseen worldwide hypertonic mannitol solution unavailability. Moreover, the treatment was interrupted in one patient after the cytoreductive MATRix cycle because of myocardial infarction. These formerly mentioned two patients were later treated with one to two conventional MATRix cycles, and they subsequently received ASCT-supported HD chemotherapy consolidation. The remaining patients (22/25) were treated per protocol.

The radiological treatment response was assessed at three timepoints: after the cytoreductive MATRix regimen, before proceeding to the HD + ASCT treatment, and at restaging. The results are presented in Table 3.

A demonstrative example of the radiologic response to the treatment is presented in Figure 1. After the MATRix regimen, 23 (92.0%) patients had a partial response to the therapy. One (4.0%) patient had a progressive disease after the MATRix regimen but became responsive to the BBBD treatment. In one patient, the response was not evaluable due to a large postoperative hemorrhage covering the residual tumor. Before the HD + ASCT treatment, five (20.0%) patients had a partial response, and nineteen (76.0%) patients had a complete response, including the patient with progressive disease after the MATRix regimen. One response was not evaluable. At restaging, two (8.0%) patients had partial and twenty-two (88.0%) had complete responses. One (4.0%) patient whose treatment was interrupted due to mannitol unavailability had experienced a relapse.

The survival rates were calculated separately for the intention-to-treat (ITT) (*n* = 25) and treated-per-protocol (PPT) populations (*n* = 22; two patients whose BBBD treatment was interrupted due to mannitol unavailability and one with cardiac issues were excluded) (Figure 2). The median follow-up time was 30 months (reverse Kaplan–Meier method). At the 2-year follow-up time point, the OS rates were 67.1% and 81.0%, PFS rates 70.3% and 81.0%, and PPT rates 77.8% and 90.5% in the ITT and PPT populations, respectively. Altogether, seven (28.0%) patients had died. Five (20.0%) patients had died of lymphoma, including three patients with early treatment interruptions. Two (8.0%) patients had died from ASCT treatment complications in complete response. No deaths during the BBBD treatment were reported. The median PFS and OS were not achieved at the 30-month follow-up period (Figure 2).

Acute hematological, neurological, and other toxicities of BBBD treatment are presented in Table 4 and toxicities from the HD + ASCT treatment in Table 4. During the BBBD treatment, severe (grades III–IV) hematological toxicities were common (Table 4). Grade III–IV anemia was reported in 80.0% of the patients, grade III–IV neutropenia in 100.0% of the patients (with grade III–IV infections in 76.0% of the patients), and grade III–IV thrombocytopenia in 92.0% of the patients. Based on hematological toxicity, dose modifications were allowed. In the study population, the mean dose reduction was 11%. The smallest reduction (8.5%) was seen with cycle 1 after cytoreductive MATRix, and the highest reduction (15%) with cycle five due to the cumulative toxicity of the treatment. However, hematological AEs were generally manageable with chemotherapy dose adjustments and well-organized supportive care. No hematological toxicity-related deaths were reported during the BBBD treatment.

The toxicities from the HD + ASCT treatment are shown in Table 4. One treatment-related death from intracerebral hemorrhage (ICH) in a thrombocytopenic HD + ASCT-treated patient and one death from late complications (>1 year) of the HD + ASCT treatment were reported.

Considering the unique BBBD technique exploited in this study, neurological toxicities were of special interest. The acute neurological toxicities are reported in Table 4. One (4.0%) patient reported grade I acute tinnitus, but hearing loss or retinopathy were not observed. In five (20.0%) patients, occult focal ischemic lesions were detected in routine follow-up MRI scans. The lesions were visible only in diffusion-weighted imaging sequences. Assessed by radiology specialists, the lesions were considered to have resulted from catheterization procedures. Considering the total of 106 treatment cycles with 212 catheterizations performed during the study, the incidence rate of the ischemic lesions was 2.4% per catheterization (5/212 catheterizations). Symptomatic CNS ischemia during BBBD therapy was not reported.

Thromboembolic events occurred in 24.0% (6/25) patients (Table 4). Other cardiovascular events were not systematically followed in the current study.

## 4. Conclusions

PCNSL has represented a continuous clinical challenge with a dismal long-term outcome [19]. Here, we report the results of a prospective phase II study evaluating the efficacy and toxicity of BBBD chemoimmunotherapy in conjunction with MATRix induction and ASCT consolidation in 25 patients with previously untreated PCNSL. We found a complete response rate of 88.0% to the treatment. A promising 2-year OS rate of 81.0%, PFS rate of 81.0%, and FFP of 90.5% among the PPT population were observed.

A major advantage of the BBBD technique in comparison to conventional intravenous dosing of chemotherapeutic agents is the increased drug delivery to CNS [18]. We have intensified the original BBBD protocol developed by Angelov et al. by including a cytoreductive MATRix cycle induction and an HD + ASCT consolidation in the protocol [18,19]. The results from the current study are congruent with our previous retrospective analysis, where the intensified regimen seemed to improve treatment efficacy [19]. Our 2-year PFS rate of 70.3% in the ITT population (81% in the PPT population) parallels the 2-year PFS rate of 61% in the IELSG-32 study MATRix arm. [14]. It is notable, however, that our study setting together with the limited sample size leave room for speculations when comparing the results with other studies. Importantly, our selection criteria allowed the inclusion of patients with the ECOG performance status of >2 when due to lymphoma symptoms. In total, 24.0% (6/25) of the patients were of ECOG 3 (Table 2). Despite the potentially deteriorating effect on the survival outcomes, we consider this extension important in the aim of representing real-world PCNSL patients.

In this trial, we observed a high rate of acute grade III–IV hematological toxicity during the BBBD and the HD+ASCT treatment (Table 4) with a treatment-related mortality of 8.0%. Hematological toxicities led to the adjustment of BBBD chemotherapy doses for most of the patients, especially for those over the age of 65 years. High treatment-related toxicity and mortality are common to most other first-line PCNSL chemotherapy regimens, and our results are consistent with them [14,15,21,24]. Whether smaller doses would suffice for a comparable efficacy requires further investigations.

The cumulative incidence of ischemic CNS events during BBBD therapy was 20%. Notably, all these lesions were small occult findings detected only at follow-up MRI without clinical sequelae. Estimated by radiology specialists, the lesions appeared to be related to catheterization procedures based on their location and timing. The incidence rate of ischemic CNS events per single catheterization procedure was low at 2.4%. Generally, the incidence rates of ischemic CNS events after neuro-interventional procedures and cerebral angiographies have been reported to be considerably higher, between 26% and 46% [25,26,27]. In the future, we aim to uncover the etiology and risk factors for this finding to assess its clinical significance more profoundly. Other neurological events were rare in this trial. One patient reported acute tinnitus, which was eventually resolved. No hearing loss was observed. Retinopathies were not noted either; however, this result may be an underestimation because our study scheme did not include a systematic ophthalmological surveillance of the patients not presenting with intraocular disease or visual symptoms.

The incidence of venous thromboembolic (VTE) complications in the study was 24%. This is not surprising considering that PCNSL patients are often highly morbid with long immobilized periods due to poor PS. High incidence of VTEs has also been reported during other PCNSL therapy protocols [28]. We later added prophylactic low molecular weight heparin treatment to the treatment protocol of our patients. We consider that the myocardial infarction that occurred in one patient was attributed to the deterioration of a pre-existing cardiovascular disease during the HD methotrexate-containing chemotherapy combined with intense intravenous hydration, rather than the BBBD procedure itself.

Our treatment protocol included one course of cytoreductive intravenous MATRix regimen before proceeding to the BBBD treatment. There are two reasons for this approach. Firstly, patients with high tumor loads tend to recover slowly from their first BBBD cycle without preceding intravenous cytoreduction. With MATRix induction, we could avoid the prolonged recovery period. Secondly, because PCNSL is a highly proliferating malignancy causing rapidly developing and potentially permanent neurological defects, urgent therapy initiation is needed to optimize neurological recovery. This is challenging considering the logistical aspects of gathering the multidisciplinary team for the BBBD treatment. The conventional therapy initiation provides a sufficient 3-week period for BBBD scheduling and preparation while enabling prompt treatment initiation. In this study, all but one patient responded to the MATRix regimen. With one patient primarily refractory to the MATRix regimen, a unique treatment decision was made to proceed to BBBD-based immunochemotherapy. The patient achieved complete response with the BBBD treatment and remained disease-free at follow-up.

Several open questions regarding the therapy protocol optimization exist. Our adjustments to the protocol were adopted from the current standard of PCNSL care and our previous experience and adapted to the restrictions imposed by local anesthesiology resources. We acknowledge the shortcomings of the protocol design, and future studies are needed to optimize the protocol. In our experience, patients interrupting their therapy after one to three courses tend to have an increased risk of relapse; however, whether variation between four and six treatment cycles results in different outcomes remains open. Moreover, the additional impact of HD + ASCT consolidation on treatment efficacy remains undetermined, and whether intravitreal therapies add extra value in the case of intraocular lymphoma manifestation remains unclear.

PCNSL may relapse several years, or even decades, after primary treatment [16]. In this regard, long-term follow-up is needed before drawing definite conclusions about the treatment efficacy. Our median follow-up time of 30 months is still short. However, our previous experience emphasizes that late relapses are rare after BBBD treatment [19]. In this series, a patient with an ocular disease at diagnosis developed late relapse 41.5 months after the treatment initiation. Intravitreal opacities were discovered soon after the end of the treatment; however, it was not until the third vitrectomy sample that the intravitreal disease relapse was verified. Longer follow-up periods will improve our understanding of this issue.

With the currently available treatment strategies, we regard PCNSL as a disease with an option for a permanent cure. A significant issue, however, is the long-term toxicity, and especially the long-term cognitive performance. Of the current PCNSL therapy modalities, WBRT appears to have the highest long-term neurotoxicity [29,30,31,32,33,34,35,36]. In previous studies, BBBD treatment has not been reported to induce long-term neurotoxicity [17,27]. Considering the modifications to the original treatment scheme in this study, it is crucial to follow the cognitive function in surviving patients. Our research protocol includes neurocognitive surveillance performed up to 5-years of follow-up and will be presented later.

Our phase II results are promising; however, the limited number of patients leaves room for interpretation of the results. Consequently, we have decided to expand our trial and continue accrual with the aim of 50 first-line patients. Outside this trial, we have the experience of treating patients up to the age of 76 years using the current regimen with dose reductions. With this background, we have also planned a trial of BBBD induction chemotherapy with a reduced dose level followed by HD + ASCT consolidation for patients between the ages of 70 and 76 years. In the future, prospective phase III multicenter trials are needed to validate our findings.

## Figures and Tables

**Figure 1 cancers-15-01341-f001:**
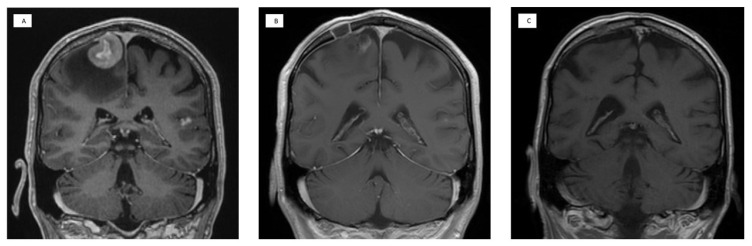
A demonstrative example of radiographic response to MATRix induction and BBBD therapy. A 64-year-old male had progressive weakness of left extremities. (**A**) Coronal contrast enhancer T1-weighted MRI revealed a hypervascular tumor in the right frontal lobe with surrounding oedema. Another hypervascular tumor was detected in the left temporal lobe. The PCNSL diagnosis was confirmed by biopsy. (**B**) Follow-up MRI after the MATRix induction showed significant debulking of both tumors. (**C**) After 4 BBBD cycles, both tumors had completely disappeared in MRI.

**Figure 2 cancers-15-01341-f002:**
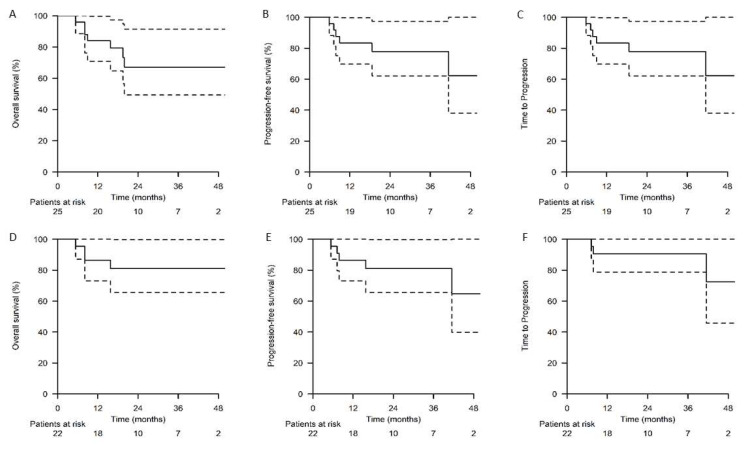
Survival outcomes. Overall survival, progression-free survival, and time-to-progression in the (**A**–**C**) ITT and (**D**–**F**) PPT populations.

**Table 1 cancers-15-01341-t001:** Chemotherapeutic agents, doses, routes, and days of administration in the BBBD ^a^ protocol.

Chemotherapy Agent	Dose	Route	Day of Cycle
Rituximab	375 mg/m^2^	intravenous	0
Methotrexate	2500 mg/m^2^	intra-arterial	1–2
Carboplatin	200 mg/m^2^	intra-arterial	1–2
Dexamethasone	6 mg × 4–6	peroral	2–10
Cytarabine	40 mg	intrathecal	14
Cyclophosphamide	330 mg/m^2^	intravenous	1–2
Etoposide	200 mg/m^2^	intravenous	1–2

^a^ BBBD = blood–brain barrier disruption.

**Table 2 cancers-15-01341-t002:** Basic demographics.

Sex	N (%)
Male	12 (48)
Female	13 (52)
**Age at diagnosis**	**(mean + SD)**
Years	61.3 ± 10.9
**Performance status (WHO ^a^)**	**N (%)**
0	4 (16)
1	10 (40)
2	5 (20)
3	6 (24)
**MSKCC ^b^ risk group**	**N (%)**
0 (age < 50 years)	4 (16)
1 (age ≥ 50 years and KPS ^c^ ≥ 70%)	11 (44)
2 (age ≥ 50 years and KPS ^c^ < 70%)	10 (40)
**Deep brain structure involvement**	**N (%)**
Yes	24 (96)
No	1 (4)
**Eye involvement**	**N (%)**
No	22 (88)
Yes	3 (12)
**Spinal involvement**	**N (%)**
No	23 (92)
Yes	1 (4)
N/A	1 (4)

^a^ WHO = World Health Organization. ^b^ MSKCC = Memorial Sloan–Kettering Cancer Center risk groups. ^c^ KPS = Karnofsky prognostic score.

**Table 3 cancers-15-01341-t003:** Responses after the MATRix induction (1 cycle), after BBBD ^a^ treatment, and at restaging.

Response	After MATRix Regimen	After BBBD Treatment	At Restaging
Complete response, N (%)	0 (0)	19 (76)	22 (88)
Partial response, N (%)	23 (92)	5 (20)	2 (8)
Stable disease, N (%)	0 (0)	0 (0)	0 (0)
Progressive disease, N (%)	1 (4)	0 (0)	1 (4)
Not evaluable, N (%)	1 (4)	1 (4)	0 (0)

^a^ BBBD = blood–brain barrier disruption.

**Table 4 cancers-15-01341-t004:** Frequency of hematological, neurological or other adverse events of the blood–brain barrier disruption treatment and toxicities from the autologous stem cell transplantation-supported high-dose treatment.

Hematological AE ^a^	All Grades, N (%)	Grade III–IV, N (%)
Anemia	25 (100)	20 (80)
Neutropenia	25 (100)	25 (100)
WBC ^b^ decreased	25 (100)	23 (92)
Platelet level decreased	25 (100)	23 (92)
Infection	21 (84)	19 (76)
Red blood cell transfusion	21 (84)	
Platelet transfusion	17 (68)	
**Neurological AE ^a^**		
Tinnitus	1 (4)	
Hearing loss	0 (0)	
Retinopathy	0 (0)	
CNS ^c^ ischemia (asymptomatic)	5 (20)	
CNS ^c^ ischemia (symptomatic)	0 (0)	
**Other AE ^a^**		
Mucositis	16 (64)	5 (20)
Nausea	17 (68)	3 (12)
Deep vein thrombosis	2 (8)	
Pulmonary embolism	4 (16)	
Osteoporotic fracture	2 (8)	
**High-dose treatment AE ^a^**		
Infection	25 (100)	25 (100)
Mucositis	23 (92)	22 (88)
Red blood cell transfusion	19 (76)	
Platelet transfusion	25 (100)	

^a^ AE = adverse event. ^b^ WBC = white blood cells. ^c^ CNS = central nervous system.

## Data Availability

The data presented in this study are available on request from the corresponding author. The data are not publicly available due to privacy and ethical restrictions.

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
