# Peer review of "Blood–Brain Barrier Disruption (BBBD)-Based Immunochemotherapy for Primary Central Nervous System Lymphoma (PCNSL), Early Results of a Phase II Study"

_cancers, 2023, doi:10.3390/cancers15041341_

Round 1
Reviewer 1 Report
Kuitunen et al report on the use of a Blood-Brain Barrier Disruption-based chemotherapy for primary central nervous system lymphoma, in 25 patients at first line treatment. The study is well described and the efficacy of this new treatment program seems interesting (CRR at final restaging 88%, 2-y OS 67% and 2-y PFS 70%). The toxicity profile is not negligible, in particular the authors describe a high rate of grade III-IV hematological toxicity with 76% of moderate-severe infections; the incidence of thromboembolic complications is high and deserves further evaluation, as well as the minor ischemic events related to the catheterization procedures, although asymptomatic. For this reason, and given the limited number of patients included, the manuscript needs to be tempered in its main conclusion.
I suggest some minor revisions of the text and references:
-ABSTRACT: line 40, the sentence "with consolidated high-dose treatment followed by autologous..." should be changed into "with high-dose treatment with autologous stem cell transplanatation as consolidation". "MATRIx" regimen needs to be corrected in "MATRix" in all the text.
-INTRODUCTION: in the first paragraph the references are not precise. In particular, the sentence about the epidemiology and increasing incidence of PCNSL has to be connected only to references n.1 and 2. After the reference n. 9 about histology (correct), the references n. 10 and 11 are lacking and the authors report directly the reference n. 12. In this second sentence about the conventional treatment of PCNSL, I recommend to cite not only the MATRix study (n.12 and 13) but also the references n. 3-4-5-6-7-8-10-11. I suggest to add some reviews and guidelines regarding PCNSL therapy, for example "Guidelines for the diagnosis and management of primary central nervous system diffuse large B-cell lymphoma" Fox C P et al Br J Hematol 2019; "Central nervous system lymphoma" Schaff LR et al Blood 2022; "European Association of Neuro-Oncology (EANO) guidelines for treatment of primary central nervous system lymphoma" Hoang-Xuan K et al Neuro Oncol 2023.
Line 78: please specify the acronym BBBD
Lines 85-87: I suggest a better description of the seminal paper by Angelov et al: how many patients? study design?
Line 90: "for which reason" needs to be corrected in "for this reason"
Line 95: "IESLG protocol" needs to be corrected in "IELSG protocol"
-MATERIALS AND METHODS, OUTCOMES: line 135, according to the results reported, I suppose the primary end point was the OS at 2 years; the same for the PFS and the TTP as secondary end points.
Line 147: I suggest to specify that the infusion of chemotherapy as in Table 1 was preceded by an intra-arterial infusion of hypertonic mannitol solution (please report the amount and concentration of this solution).
-RESULTS: line 206, after how many cycles did this patient with myocardial infarct interrupt the treatment program?
Line 219: Please specify if all but one patients proceded to ASCT
Line 256: Please specify the acronym ICH
CONCLUSIONS: line 285 "PCNSL has represented" instead of "has presented"; line 306 "a treatment-related mortality of 8.0%" instead of "the treatment-related mortality of 8.0%".
The authors state that the hematological toxicity lead to an adjustment of BBBD chemotherapy doses for most patients but do not report details. I would recommend a specific paragraph with dose reductions in "RESULTS".
Given the high rate of hematological toxicity/infections and 8% of treatment-related mortality, the authors may suggest further studies in order to evaluate the effects of a dose reduction. I would suggest to delete the statements at lines 311-312 "we consider that the high rate of acute hematological toxicity is justified because it is manageable and enables the patients to achieve long-term remission" and at lines 370-372 " we believe that even severe acute treatment-related toxicity could be considered acceptable if they are inevitable for achieving good disease control".
Author Response
Reviewer 1
Kuitunen et al report on the use of a Blood-Brain Barrier Disruption-based chemotherapy for primary central nervous system lymphoma, in 25 patients at first line treatment. The study is well described and the efficacy of this new treatment program seems interesting (CRR at final restaging 88%, 2-y OS 67% and 2-y PFS 70%). The toxicity profile is not negligible, in particular the authors describe a high rate of grade III-IV hematological toxicity with 76% of moderate-severe infections; the incidence of thromboembolic complications is high and deserves further evaluation, as well as the minor ischemic events related to the catheterization procedures, although asymptomatic. For this reason, and given the limited number of patients included, the manuscript needs to be tempered in its main conclusion.
I suggest some minor revisions of the text and references:
ABSTRACT: line 40, the sentence "with consolidated high-dose treatment followed by autologous..." should be changed into "with high-dose treatment with autologous stem cell transplanatation as consolidation". "MATRIx" regimen needs to be corrected in "MATRix" in all the text.
As suggested, we have now revised the term “MATRIx” to “MATRix” thoroughly the text.
INTRODUCTION: in the first paragraph the references are not precise. In particular, the sentence about the epidemiology and increasing incidence of PCNSL has to be connected only to references n.1 and 2. After the reference n. 9 about histology (correct), the references n. 10 and 11 are lacking and the authors report directly the reference n. 12. In this second sentence about the conventional treatment of PCNSL, I recommend to cite not only the MATRix study (n.12 and 13) but also the references n. 3-4-5-6-7-8-10-11. I suggest to add some reviews and guidelines regarding PCNSL therapy, for example "Guidelines for the diagnosis and management of primary central nervous system diffuse large B-cell lymphoma" Fox C P et al Br J Hematol 2019; "Central nervous system lymphoma" Schaff LR et al Blood 2022; "European Association of Neuro-Oncology (EANO) guidelines for treatment of primary central nervous system lymphoma" Hoang-Xuan K et al Neuro Oncol 2023.
As suggested, we have revised references, and added two references to the manuscript (Hoang-Xuan et al. 2023 and Schaff et al. 2022).
Line 78: please specify the acronym BBBD.
As suggested, we have specified the acronym BBBD. Please see line 76.
Lines 85-87: I suggest a better description of the seminal paper by Angelov et al: how many patients? study design?
We have added the following sentences to the manuscript: “In the study 149 newly diagnosed PCNSL patients were treated with 12 chemotherapy courses without consolidation therapy with BBBD and intra-arterial (IA) methotrexate. The study demonstrated a good safety profile and neurocognitive tolerance. The overall response rate was 81.9% and median overall survival was 3.1 years.” Please see lines 84-88.
Line 90: "for which reason" needs to be corrected in "for this reason"
Revised as suggested. Please see line 91.
Line 95: "IESLG protocol" needs to be corrected in "IELSG protocol"
Revised as suggested. Please see line 96.
MATERIALS AND METHODS, OUTCOMES: line 135, according to the results reported, I suppose the primary end point was the OS at 2 years; the same for the PFS and the TTP as secondary end points.
The primary and secondary end points are 2, 5 and 10 years OS, PFS and TTP. Since the number of patients is still low or absent at the follow-up points 5 and 10, we are able to report only 2 year OS, PFS, and TTP.
Line 147: I suggest to specify that the infusion of chemotherapy as in Table 1 was preceded by an intra-arterial infusion of hypertonic mannitol solution (please report the amount and concentration of this solution).
We have added the following sentences to the manuscript. “In the treatment schedule prior to the first or second treatment course, a porta-cath was inserted and patients were hydrated at 100–150 mL/h for a minimum of 6 h. Because of the risk of seizures patients were premedicated with an anticonvulsant. Rituximab was administered on day 0. On day 1 and 2, Atropine was administered intravenously immediately prior to BBBD to prevent bradycardia. BBB opening was conducted under general anaesthesia. Internal carotid artery (ICA) or vertebral artery (VA) was selectively catheterised via transfemoral access. In ICA, the catheter tip was placed at C1–C2 level and in the vertebral artery at C6 level. Cyclophoshamide and etoposide are pro-drugs and activate liver metabolism and were administered intra-venously before mannitol infusion. Warmed mannitol (25 %) was administered at 4–6 mL/s into the target and immediately after BBBD, methotrexate and carboplatin were administered intra-arterially. Following BBBD, patients remained in the post-anaesthesia care unit with frequent monitoring of vital signs, neurologic status, and fluid balance. Fluid balance was meticulously maintained with mannitol (15%) and fluid boluses. In patients treated with methotrexate, NaHCO3 was added to intravenous fluids and titrated to achieve urine pH greater than 7.5.” Please see lines 153-168.
RESULTS: line 206, after how many cycles did this patient with myocardial infarct interrupt the treatment program?
After only one cycle. We have modified the following sentence in the manuscript: “The treatment was interrupted in one patient after cytoreductive MATRix cycle because of myocardial infarction.” Please see lines 224-225.
Line 219: Please specify if all but one patients proceded to ASCT
We have modified the sentence followingly: “Moreover, the treatment was interrupted in one patient after cytoreductive MATRix cycle because of myocardial infarction. These formerly mentioned two patients were later treated with one to two conventional MATRix cycles, and they subsequently received ASCT-supported HD chemotherapy consolidation. The remaining patients (22/25) were treated per protocol.” Please see lines 225-228.
Line 256: Please specify the acronym ICH
We have specified the acronym ICH as suggested. Please see line 279.
CONCLUSIONS: line 285 "PCNSL has represented" instead of "has presented"; line 306 "a treatment-related mortality of 8.0%" instead of "the treatment-related mortality of 8.0%".
We have modified the sentences as suggested. Please see line 309.
The authors state that the hematological toxicity lead to an adjustment of BBBD chemotherapy doses for most patients but do not report details. I would recommend a specific paragraph with dose reductions in "RESULTS".
We have added the following sentences to the manuscript: “Based on hematological toxicity, dose modifications were allowed. In the study population mean dose reduction was 11%. The smallest reduction (8.5%) was seen with cycle 1 after cytoreductive MATRix and the highest reduction (15%) with cycle five due to cumulative toxicity of the treatment.” Please see lines 271-274.
Given the high rate of hematological toxicity/infections and 8% of treatment-related mortality, the authors may suggest further studies in order to evaluate the effects of a dose reduction. I would suggest to delete the statements at lines 311-312 "we consider that the high rate of acute hematological toxicity is justified because it is manageable and enables the patients to achieve long-term remission" and at lines 370-372 " we believe that even severe acute treatment-related toxicity could be considered acceptable if they are inevitable for achieving good disease control".
As suggested we have omitted these statements.

Reviewer 2 Report
BBBD is very labor intensive and feasible only at a few centers. The early data look favorable but the real question should be longer term follow up. The primary endpoints are 2-, 5-, and 10-year OS and this paper has follow up for only 2 years. Most treatments for a chemo-responsive tumor like PCNSL can look impressive early on, but the important measure of the value of such an intensive treatment is longer term OS - 5- and 10-year OS as you imply by your endpoints. As such, the title should add something like:..."Early results of a phase II study."
Also, you compare your 2-year PFS to the IELSG result of 61%. In the current data set, the 2-yr PFS is 67% for the ITT population. The 81% you mention (line 297) is for the PPT which is not a proper number to use for this comparison.
One of your secondary endpoints is "TTP rate" but the rate does not make sense since it is a time not a rate. Did you mean freedom from progression? Please clarify.
The statistics section needs to have assumptions of efficacy. For instance, the 2-yr OS assumption at the start of the trial would be something like: "The 2-yr OS of >X% would be considered a positive study."
How can the PFS rate better than OS (e.g., line 51, 237)? This is impossible - if a patient dies they are no longer progression free. Please explain.
Minor points:
"This data" (line 34) should be "These data."
Author Response
Reviewer 2
BBBD is very labor intensive and feasible only at a few centers. The early data look favorable but the real question should be longer term follow up. The primary endpoints are 2-, 5-, and 10-year OS and this paper has follow up for only 2 years. Most treatments for a chemo-responsive tumor like PCNSL can look impressive early on, but the important measure of the value of such an intensive treatment is longer term OS - 5- and 10-year OS as you imply by your endpoints. As such, the title should add something like:..."Early results of a phase II study."
As suggested, we have modified the title followingly: “Blood–Brain Barrier Disruption (BBBD)-Based Immunochemo-therapy for Primary Central Nervous System Lymphoma (PCNSL), Early Results of A Phase II Study”. Please see line 4.
Also, you compare your 2-year PFS to the IELSG result of 61%. In the current data set, the 2-yr PFS is 67% for the ITT population. The 81% you mention (line 297) is for the PPT which is not a proper number to use for this comparison.
Thank you for your valuable comment. We have modified the sentence in order to clarify this followingly: ”Our 2-year PFS rate of 70.3% in the ITT population (81% in the PPT population) parallels with the 2-year PFS rate of 61% in the IELSG-32 study MATRix arm.” Please see lines 320-322.
One of your secondary endpoints is "TTP rate" but the rate does not make sense since it is a time not a rate. Did you mean freedom from progression? Please clarify.
This is a very important comment. We apologize our imprecise terminology. Freedom from progression is a precise term. We have added the following sentence to the manuscript in order to define the term. “Freedom from progression means the proportion of patients whose disease is not progressing at the selected time and patients who died due to other causes were censored." Please see line144-146.
We have also added the term “FFP” to the outcomes. Please see line 140.
In addition we have replaced the term TTP with FFP in line 314.
The statistics section needs to have assumptions of efficacy. For instance, the 2-yr OS assumption at the start of the trial would be something like: "The 2-yr OS of >X% would be considered a positive study."
In our retrospective unpublished data we have found long-term survival rate 67% with BBBD therapy. In our previously published retrospective analyses concerning Finnish PCNSL patients treated with Bonn protocol we found 5-year PFS rate 38% (Harjama et al., 2015). Based on those outcomes we calculated that in order to be able to prove 50% increase in PFS rate, we would need 24 first-line patients. Since we have not published all the data yet, we prefer not to add the suggested sentence into the manuscript.
Reference: Harjama L, Kuitunen H, Turpeenniemi-Hujanen T, Haapasaari KM, Leppä S, Mannisto S, Karjalainen-Lindsberg ML, Lehtinen T, Eray M, Vornanen M, Haapasalo H, Soini Y, Jantunen E, Nousiainen T, Vasala K, Kuittinen O. Constant pattern of relapse in primary central nervous lymphoma patients treated with high-dose methotrexate combinations. A Finnish retrospective study. Acta Oncol. 2015 Jun;54(6):939-43. doi: 10.3109/0284186X.2014.990110. Epub 2015 Mar 11. PMID: 25761092.
How can the PFS rate better than OS (e.g., line 51, 237)? This is impossible - if a patient dies they are no longer progression free. Please explain.
In general PFS should be the same or worse than OS. However, in materials with small sample size and limited follow-up time, like ours, it may happen that the number of patients censored between events may vary in PFS and OS analyses, thus leading to situation where the event occurs among different number of patients at risk, thus leading to different proportion of patients event free. In our material, before 24 months time point, seven events occured in both PFS and OS curves (two treatment related deaths and 5 patient had relapses and all of them died to lymphoma before 24 month). Relapses naturally occurred earlier than lymphoma related deaths, when there were more patients in active follow-up, thus the degree in percentages after one event was smaller compared to OS, where more patients had been censored, leading to better 24 months event free number on PFS compared to OS.
Minor points: "This data" (line 34) should be "These data."
Modified as suggested. Please see line 34.

Reviewer 3 Report
Kuitunen et al. present results of an interesting phase II study looking at the efficacy of blood brain barrier disruption followed by intraarterial administration of methotrexate and carboplatin in conjunction with intravenous administration of rituximab, dexamethasone, cytarbine, cyclophosphamide and etoposide and subsequent ASCT-supported HD chemotherapy using BCNU and thiotepa. The results seem to be acceptable both in terms of efficacy and safety, although the efficacy was not game changing and one death during ASCT-supported HD chemotherapy is certainly concerning. I have the following questions/comments.
1. In the methods, please specify how mannitol was administered for BBBD.
2. For Table 3, it should be specified that "after MATRIx regimen" is only 1 course. The results (many PR and no CR) seem to be reasonable for after 1 course of chemotherapy.
3. It would be great if PK/PD studies could be performed in the brain (tumor) tissue after BBBD therapy. Would this be too invasive?
4. Is there any consensus as to the optimal BBBD therapy regimen for PCNSL?
Author Response
Reviewer 3
Kuitunen et al. present results of an interesting phase II study looking at the efficacy of blood brain barrier disruption followed by intraarterial administration of methotrexate and carboplatin in conjunction with intravenous administration of rituximab, dexamethasone, cytarbine, cyclophosphamide and etoposide and subsequent ASCT-supported HD chemotherapy using BCNU and thiotepa. The results seem to be acceptable both in terms of efficacy and safety, although the efficacy was not game changing and one death during ASCT-supported HD chemotherapy is certainly concerning. I have the following questions/comments.
- In the methods, please specify how mannitol was administered for BBBD.
We have added the specification of manniltol administration to the manuscript. Please see the lines 162-166.
- For Table 3, it should be specified that "after MATRIx regimen" is only 1 course. The results (many PR and no CR) seem to be reasonable for after 1 course of chemotherapy.
We have modified the title of the table 3 as following: “Responses after the MATRix induction (1 cycle), after BBBDa treatment, and at restaging.” Please see line 232.
- It would be great if PK/PD studies could be performed in the brain (tumor) tissue after BBBD therapy. Would this be too invasive?
Since the brain tissue biopsies are highly invasive, biopsies are not justified from a treatment point of view.
- Is there any consensus as to the optimal BBBD therapy regimen for PCNSL?
At the moment, there are two research group working with BBBD treatment. Therapy regimens differ from used multichemocombinations (four or five drug regimen) and administration intervals. The lack of randomized trials prevents to conclude optimal BBBD regimen.

Round 2
Reviewer 3 Report
The authors have satisfactorily answered to the suggestions raised by the reviewers. The paper should now be considered suitable for publication.